# Blocking IL-10 signaling with soluble IL-10 receptor restores *in vitro* specific lymphoproliferative response in dogs with leishmaniasis caused by *Leishmania infantum*

**Catiule de Oliveira Santos[1], Sidnei Ferro Costa[2], Fabiana Santana Souza[1], Jessica Mariane Ferreira Mendes[1], Cristiane Garboggini Melo de Pinheiro[1], Diogo Rodrigo de Magalhães Moreira[1], Luciano Kalabric Silva[1], Valeria Marçal Felix de Lima[2], Geraldo Gileno de Sá Oliveira[1] ***

**1** Oswaldo Cruz Foundation, Gonçalo Moniz Research Center, Laboratory of Structural and Molecular Pathology (LAPEM), Tissue Engineering and Immunopharmacology Laboratory (LETI) or Pathology and Molecular Biology Laboratory (LPBM), Candeal, Salvador, Brazil, **2** Department of Clinical Medicine, Surgery and Animal Reproduction, São Paulo State University (UNESP), School of Veterinary Medicine, Araçatuba, Brazil

* geraldo.oliveira@fiocruz.br, ggsoliveir@gmail.com

## Abstract

rIL-10 plays a major role in restricting exaggerated inflammatory and immune responses, thus preventing tissue damage. However, the restriction of inflammatory and immune responses by IL-10 can also favor the development and/or persistence of chronic infections or neoplasms. Dogs that succumb to canine leishmaniasis (CanL) caused by *L. infantum* develop exhaustion of T lymphocytes and are unable to mount appropriate cellular immune responses to control the infection. These animals fail to mount specific lymphoproliferative responses and produce interferon gamma and TNF-alpha that would activate macrophages and promote destruction of intracellular parasites. Blocking IL-10 signaling may contribute to the treatment of CanL. In order to obtain a tool for this blockage, the present work endeavored to identify the canine casIL-10R1 amino acid sequence, generate a recombinant baculovirus chromosome encoding this molecule, which was expressed in insect cells and subsequently purified to obtain rcasIL-10R1. In addition, rcasIL-10R1 was able to bind to homologous IL-10 and block IL-10 signaling pathway, as well as to promote lymphoproliferation in dogs with leishmaniasis caused by *L. infantum*.

## 1 Introduction

Cytokines are polypeptides that participate in communication between cells and the orchestration of immune system responses. In response to tissue damage and/or stimulation, these molecules may be secreted and bind to the extracellular domains of cognate receptors on target cells. This results in the phosphorylation of intracytoplasmic domains of cytokine receptors and activation of transcription factors, which migrate to the nucleus and implement cellular

**Data Availability Statement:** All relevant data are within the manuscript and its Supporting Information files.

**Funding:** COS, Master Degree Scholarhip, FAPESB FSS, Scientific initiation program, FAPESB SFC, JMFM, CGMP, Coordination for the Improvement of Higher Education Personnel (CAPES) - Finance Code 001 VMFL, National Council for Scientific and Technological Development (CNPq) process #400063/2016-6 and São Paulo Research Foundation (FAPESP), process #2017/10906-8. GGSO, CNPq/INCT-DT, process #573839/2008-5 and CNPq-PROEP, process #400913/2013-5 The funders had no role in study design, data collection and analysis, decision to publish, or preparation of the manuscript.

**Competing interests:** No authors have competing interests.

transcription and functions [1]. Some cytokines mainly promote pro-inflammatory activities, while others are anti-inflammatory or immunosuppressive in nature [2]. Proinflammatory cytokines are predominantly produced by activated macrophages and lymphocytes, including interleukein-1β (IL-1β), IL-6, tumor necrosis factor alpha (TNF-α), granulocyte-macrophage colony stimulating factor (GM-CSF), interferon gamma (IFN-γ) and IL-12 [1,2]. Anti-inflammatory cytokines are produced mainly by lymphocytes or non-classically activated macrophages, e.g. IL-4, IL-10, IL-11, IL-13 and TGF-β [2–4].

IL-10, the most important anti-inflammatory and immunosuppressive cytokine, can be produced by a variety of immune cells, including CD4+ and CD8+ T cells, B lymphocytes, natural killer (NK) cells, monocytic and dendritic cells, as well as eosinophils and neutrophils [5,6]. IL-10 signaling occurs via a receptor consisting of two distinct polypeptide chains, subunits IL-10Rα (IL-10R1) and IL-10Rβ (IL-10R2). Accordingly, IL-10 initially binds to the extracytoplasmic domain (ECD) of IL-10R1 with high affinity, followed by low affinity interactions with the IL-10R2 subunit by both IL-10 and IL-10R1 [7]. Next, JAK1 and TYK2 respectively interact with IL-10R1 and IL-10R2, become phosphorylated and mainly activate STAT3 [8], leading to the implementation of gene transcription programs and consequent cellular responses [5,6,8].

IL-10 can target several cells in the immune system and may exert a broad range of anti-inflammatory and immunosuppressive activities on these cells. As a result of high IL-10R expression, monocytes and macrophages are the main targets for the inhibitory effects of IL-10 [9–11]. In these cells, IL-10 inhibits the transcription of cytokines and chemokines (IL-1α, IL-1β, IL-6, IL-10, IL-12, IL-18, GM-CSF, G-CSF, M-CSF, TNF-α, LIF and PAF) [12–15] and reduces antigen presentation by decreasing the expression of MHCII, accessory (CD86) and adhesion (CD54) molecules [16–18]. Although IL-10 increases phagocytic activity in macrophages [19,20], it limits the production of superoxide anion ($O_2^-$) and nitric oxide (NO) in these cells, hampering the elimination of phagocytosed microorganisms [21–23]. In addition, IL-10 also inhibits some T lymphocyte functions indirectly through decreased antigen presentation [24] and directly by inhibiting CD4 T cell proliferation and cytokine production (IL-2 and IFN-γ) [25,26]. However, IL-10 exerts stimulatory effects over CD8$^+$ T cells, inducing recruitment, proliferation and cytotoxic activity [27–29].

Canine leishmaniasis (CanL) caused by *Leishmania infantum* (synonymous with *L. chagasi* in the Americas) is a serious disease caused by the obligate intracellular protozoan [30,31]. Following natural inoculation with *L. infantum*, dogs may or may not develop disease [32]. Those that are susceptible may present mild signs or even develop severe and fatal disease [32,33]. Dogs that develop the symptomatic form of leishmaniasis may exhibit higher IL-10 and lower IFN-γ concentrations in the blood, while the inverse it true in asymptomatic dogs [34,35]. A positive correlation is evidenced between the expression of IL-10 and *L. infantum* parasitic load in the lymph nodes and spleens of infected canines. Under stimulation with leishmania antigens, susceptible animals exhibit an inability to mount a cellular immune response, as evidenced by the lack of lymphoproliferative response and cytokine production (IFN-γ and TNF-α), which stimulates microbicidal mechanisms in macrophages. The addition of IL-10 to cultures of peripheral blood mononuclear cells (PBMC) has been shown to inhibit the lymphoproliferative response to leishmania antigens [36].

The immunization of animals with antigens concomitantly with the blocking of IL-10 signaling may favor the induction of a cellular immune response (Th1), even in the course of infection. This may represent a valid strategy in the development of preventive or therapeutic vaccines, as well as immunotherapeutic protocols, against canine diseases, including leishmaniasis caused by *L. infantum* [37–40].

The present work aimed to manipulate immune responses in dogs by producing recombinant casIL-10R1 (rcasIL-10R1) in a baculovirus-insect cell system, and evaluated this generated molecule's ability to bind to IL-10 inhibit signaling, and restore a lymphoproliferative response in dogs with leishmaniasis.

## 2. Material and methods

### 2.1 DNA construct, recombinant baculovirus and protein expression

Initially, the amino acid sequence of the extra-cytoplasmic domain of canine IL-10 receptor alpha chain (R1) (so-called soluble canine IL-10 receptor, casIL-10R1) was identified. This was performed by comparing the amino acid sequences of human (huIL-10R1, GeneBank, accession number NM_001558) and canine (caIL-10R1, Genbank accession number XM_005620306.1) IL-10 receptor alpha chain, and husIL-10R1 [41] using Basic Local Align Search Tool (BLAST, https://blast.ncbi.nlm.nih.gov/Blast.cgi), as well as defining transmembrane domains using an online tool https://tmdas.bioinfo.se/DAS/. After that, the DNA construct (*GP64-casIL-10R1-6H*) was designed to encode the following elements in tandem: a) *Autographa californica* multiple nuclear polyhedrosis virus (AcMNPV) GP64 leader sequence [42], b) casIL-10R1, and c) six histidines. Restriction endonuclease sites for *Sal*I and *Not*I were added to the 5' and 3' ends, respectively, of the construct. The *GP64-casIL-10R1-6H* construct was synthesized using codons optimized for expression in insect cells, then cloned into a *pUC57-Kan* plasmid (GenScript, Piscataway, USA), generating the *pUC57-Kan-GP64-casIL-10R1-6H* construct. This DNA construct was then transferred into a *pFastBac1* plasmid after digestion with *Sal*I and *Not*I and the use of T4 ligase, resulting in the *pFastBac1-GP64-casIL-10R1-6H* plasmid construct. Next, the DNA segment between Tn7R and Tn7L of *pFastBac1-GP64-casIL-10R1-6H* was transposed into a baculovirus artificial chromosome using *Escherichia coli* (DH10Bac-AcBacΔCC), as previously described [43]. Recombinant bacmid (*AcBacΔcc-GP64-casIL-10R1-6H*) was purified from *E. coli* and the presence of the insert was confirmed by PCR using primers (*5′-GTTTTCCCAGTCACGAC* and *5'-CAGGAAACAGC-TATGAC)*. This recombinant bacmid was used to generate viral stocks, which were titrated by limiting dilutions using Sf9 cells [43]. The optimization of recombinant protein expression (rcasIL10R1) was performed in High-Five cells and evaluated by dot-blotting as previously described [43].

### 2.2 Recombinant protein production

Rcas-IL10R1 was produced following a previously described method [43]. Briefly, High Five cells were cultured in Express-Five SFM medium supplemented with L-glutamine, grown to exponential phase, and then infected with the recombinant baculovirus (*AcBacΔcc-GP64-casIL-10R1-6H)* with a multiplicity of infection (MOI) of 5 for 72 hours (TOI 72 h). The cell suspension was centrifuged at 3,000 x g for 15 minutes at 4˚C to remove cell debris, and the supernatant was spun down at 30,000 x g for 1 hour at 4˚C. The resulting supernatant was stored at -70˚C until use. For purification, the thawed supernatant was dialyzed against PBS with 30 mM imidazole, pH 7.2 (binding buffer) and applied to a Sepharose-Nickel column (HisTrap HP, General Eletrics Healthcare) equilibrated with binding buffer. A HisTrap column was eluted with PBS-500 mM imidazole, pH 7.2. After analysis by SDS-PAGE, the chromatographic fractions containing rcasIL-10R1 were pooled together, submitted to dialysis against PBS, aliquoted and stored at -70˚C until use. Protein concentrations were determined by Micro BCA (Thermo Fisher Scientific, Rockford, USA). Endotoxin concentration was determined using Limulus Amebocyte Lysate (Gel-clot Method, Pyrotell, USA) [44]. Purified

recombinant protein was confirmed by Western blot assay using anti-his antibodies, as previously described [43].

## 2.3 Binding of rcasIL-10R1 to IL-10

Purified rcasIL-10R1 was immobilized on a CM5 chip (General Electrics, Uppsala, Sweden) in a Biacore T100 analyzer (General Electrics) in accordance with the manufacturer's recommendations. Briefly, 800 μL of rcasIL-10R1 (0.5 μg/mL) was applied (50 μL/minute) to a chip matrix activated by 1-Ethyl-3-(3-dimethylaminopropyl) carbodiimide hydrochloride (EDC)/ N-Hydroxysuccinimide (NHS) to achieve 1000 resonance response units (RU). The remaining reactive chemical groups on the chip were blocked by applying Ethanolamine hydrochloride-NaOH for one minute. Next, the following samples were applied to the matrix for two minutes and 30 seconds: a) phosphate buffered saline (PBS) containing 1% bovine serum albumin (BSA) and 0.05% Tween 20; b) canine IL-4 (R&D Systems, Minneapolis, USA) at 125, 250 or 500 ng/mL; c) canine IL-10 (R&D Systems) at 31.2, 62.5, 125, 250 or 500 ng/ml. IL-4 and IL-10 were diluted with PBS containing 0.05% Tween 20. After each analyte binding evaluation, matrix regeneration was performed by applying regeneration buffer for 30 seconds. Each sample was evaluated twice and results are presented as means and standard deviations (X±SD) of RU. Kinetics of association (Kon) and dissociation (Koff), as well as affinity of rcasIL-10R1 to IL-10, were calculated from RU measurements using Origin v.8.5 software (OriginLab Corporation, Northampton, USA).

## 2.4 Murine mast cell (MC/9 cell) proliferation

The murine mast cell line MC/9 (ATCC CRL-8306) was maintained by following the manufacturer's recommendations. MC/9 cell culturing was carried out in complete DMEM (Dulbecco's Modified Eagle's Medium supplemented with 10% fetal bovine serum (FBS), 2 mM L-glutamine, 0.05 mM 2-mercaptoethanol, sodium bicarbonate at 1.5 g/L and 10% of supernatant from concanavalin A (Con-A)-stimulated rat splenocytes). MC/9 cells were washed in DMEM and adjusted to $2 \times 10^5$ cells/mL in complete DMEM. The cell suspension was placed in triplicate wells (50 μL/well) on a 96-well flat-bottomed microtiter plate. One of the following solutions (50 μL/well) containing complete DMEM was added to each well: a) complete DMEM alone (negative control); b) 1.25% of supernatant from Con-A-stimulated rat splenocytes (assay positive control), c) recombinant canine IL-4 (rcaIL-4, 360 ng/mL) (R&D Systems), d) recombinant canine IL-10 (rcaIL-10, 40 ng/mL) (R&D Systems), e) rcaIL-4 (360 ng/mL) and rcaIL-10 (40 ng/mL), or f) rcaIL-4 (360 ng/mL), rcaIL-10 (40 ng/mL) and rcasIL-10R1 (8 μg/mL). The plate was placed in a humidified atmosphere for 48 h under 5% $CO_2$ at 37˚C. Then, 10 μL of Alamar Blue (Invitrogen, Carlsbad, USA) were added to each well. Cells were cultured for an additional 24 hours and optical density (OD) was read at 570 nm and 600 nm wavelengths. Differences in mean OD values were used to estimate MC/9 cell proliferation rates. Data analysis was performed using GraphPad Prism software (GraphPad Software, Inc., La Jolla, CA, USA) version 6.0.

## 2.5 Animals

This study was approved by the Brazilian Society of Science on Laboratory Animals/Brazilian College of Animal Experimentation (SBCAL/COBEA), and the Committee for Animal Care and Use–São Paulo State University (UNESP), protocol no. 00765–2017. The approved license covered the use of 5 healthy and 10 diseased dogs. A previous report characterized these animals, including clinical data [45]. Out of the 15 dogs, five were healthy (negative controls, two males and three females, two mongrels, one blue heeler, one cocker spaniel and one golden

retriever) and 10 (CanL) were diagnosed with leishmaniasis (*Leishmania infantum)* (six males and four females, seven mongrels, two poodles and one blue heeler). All control dogs tested negative for *Leishmania* DNA and *Leishmania*-specific antibodies by real-time PCR and ELISA, respectively, and presented complete blood counts and mean serum biochemistry parameters within reference ranges [45]. The 10 CanL dogs selected from the Araçatuba Zoonosis Control Center presented at least three of the following characteristic clinical signs of leishmaniasis: onychogryphosis, cachexia, ear-tip injuries, periocular lesions, alopecia, skin lesions or lymphadenopathy. *Leishmania DNA* was detected in the peripheral blood of each diseased dog by real-time PCR [45].

## 2.6 Lymphoproliferation assay

A lymphoproliferation assay was carried out as previously described [45]. Briefly, peripheral blood samples from both groups (controls and CanL) were collected in EDTA tubes. Peripheral blood mononuclear cells (PBMCs) were isolated by gradient centrifugation using Histopaque 1077 (Sigma, USA) according to the manufacturer's recommendations. Isolated cells were then washed in PBS (pH 7.2) and suspended in RPMI 1640 supplemented with inactivated 10% FBS, 0.03% L-glutamine, 100 IU/mL penicillin and 100 mg/mL streptomycin. PBMCs were stained with carboxyfluorescein diacetate succinimidyl ester (2.5 μM) (CFSE, CellTrace, Invitrogen, UK) for 10 min at 37 $^o$C in accordance with manufacturer recommendations. Stained PBMCs were cultured on sterile 96-well plates ($1 \times 10^6$/mL) with one of the following solutions containing RPMI 1640 medium: a) RPMI 1640 medium alone (negative control), b) 20 μg/mL of soluble leishmania antigens (SLA) (MHOM/BR/00/MERO2), c) rcasIL-10R1 (4 μg/mL), d) rcasIL-10R1 in the presence of SLA (20 μg/mL), e) phytohemagglutinin-M (PHA-M, 5 μL/mL) (positive control). Plates were cultured for 5 days under 5% $CO_2$ at 37 $^o$C. Events (10,000) were acquired on a flow cytometer (BD C5 Accuri Flow Cytometer, USA) and data analysis was performed using BD Accuri C6 software, v. 1.0 (BD Biosciences, CA, USA). Cell populations of similar size and complexity as the lymphocyte population were gated and evaluated by positive CFSE labeling.

## 2.6 Statistical analysis

Statistical analysis was performed using GraphPad Prism v6 software (GraphPad Software, Inc., La Jolla, CA, USA). Statistical variables were tested for normality using the Shapiro-Wilk test. Friedman's test with Dunn's post-test was used to compare lymphoproliferation rates. The Mann-Whitney test was used to compare results among groups. Values were considered significant when $p < 0.05$.

## 3. Results

### 3.1 Cloning and production of rcasIL-10R1

Initially, the amino acid sequence of the extra-cytoplasmic domain of canine IL-10 receptor alpha chain (R1) (casIL-10R1) was identified. This was carried out by comparing the entire canine IL-10R1 predicted protein (Genbank accession number XM_005620306.1), full-length human IL-10R1 (GeneBank, accession number NM_001558) and, human IL-10R1 extra-cytoplasmic domain [41] using the Basic Local Align Search Tool (BLAST, https://blast.ncbi.nlm.nih.gov/Blast.cgi). In addition, canine IL-10R1 extra-cytoplasmic domain was confirmed by analyzing full-length human and canine IL-10R1 using an online tool https://tmdas.bioinfo.se/DAS/. This analysis revealed all domains (protein signal peptide, extra-cytoplasmic, transmembrane, intracytoplasmic) of the human protein and allowed us to infer the corresponding domains in the canine sequence (Fig 1), therefore, allowing identification of canine extra-

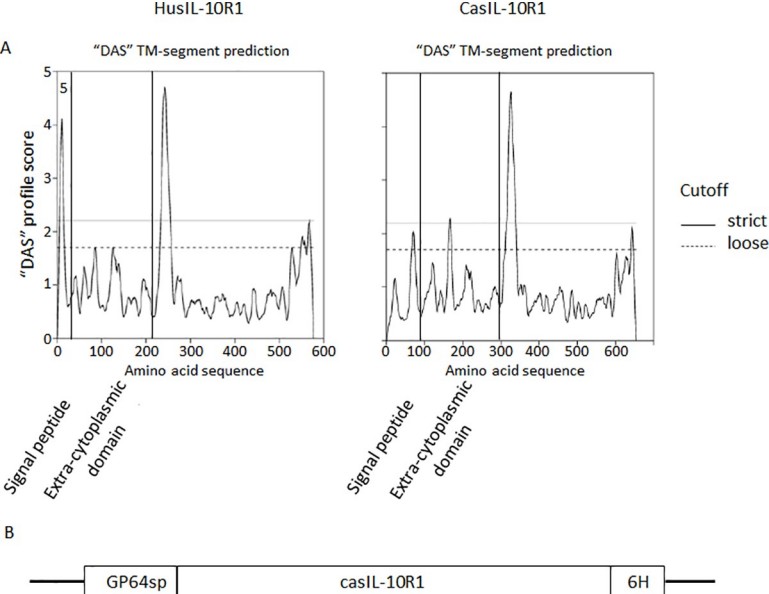

**Fig 1. Identification and design of DNA construct encoding soluble canine IL-10R1 production.** Identification of the extra-cytoplasmic domain of canine IL-10 receptor alpha chain was performed by comparing the amino acid sequences of human (huIL-10R1, GeneBank, accession number NM_001558) and canine (caIL-10R1, Genbank accession number XM_005620306.1) IL-10 receptor alpha chain, and husIL-10R1, previously described by Tan et al. 1991, using Basic Local Align Search Tool, as well as defining signal peptide and transmembrane domains (TM) (defined by vertical bars) using an online tool (https://tmdas.bioinfo.se/DAS/) (A). The DNA construct was designed to encode the following elements in tandem: a) Autographa californica multiple nuclear polyhedrosis virus (AcMNPV) GP64 leader sequence, b) casIL-10R1, and c) six histidines (B).

cytoplasmic domain. CasIL-10R1 was defined as an array of 215 amino acids which exhibited 74% similarity to homologous region of human protein. Molecular weight and isoelectric point predicted for mature rcasIL-10R1 were 25.7 kDa and PI 8.3, respectively. A DNA construct was synthesized to encode in tandem the AcMNPV GP64 signal peptide, casIL-10R1, and a 6-histidine tag. RcasIL-10R1 was purified from supernatant of High-five cells infected with the recombinant baculovirus by affinity chromatography. Purified rcasIL-10R1 showed one strong and two weak bands of 40, 28 and 23 kDa in SDS-PAGE, respectively (Fig 2A), however only the highest band was detected using anti-histidine antibodies by Western blotting (Fig 2B). The yield of the purified rcasIL-10R1 was 2.8 mg/L of High-five cell culture and the endotoxin concentration was less than 0.03 EU/mg of protein. Together, these data indicate that the recombinant protein was successfully produced.

### 3.2 Evaluation of binding between rcasIL-10R1 and canine IL-10

To assess binding between canine rcasIL-10R1 and IL-10, rcasIL-10R1 was covalently immobilized to carboxymethylated dextran matrix (CM5 chip) activated by EDC/NHS to achieve 1000 resonance response units (RU) in a Biacore T100 device. After blocking the remaining reactive chemical groups on the matrix, two samples of either PBS containing 1% BSA and 0.05% Tween 20 (to determine the baseline signal), various concentrations of canine IL-4 (cytokine irrelevant to the system, negative control) or various canine concentrations of IL-10, were applied on the matrix and RU readings were taken, and X ± SD of RU were calculated from them. When samples of PBS containing 1% BSA and 0.05% Tween 20 or IL-4 in concentrations of 125, 250 or 500 ng/ml were applied to the matrix, $1.0 \pm 1.2$, $17 \pm 6$, $15.1 \pm 4.4$, and $510 \pm 132$ RU were observed, respectively (Fig 3). On the other hand, when samples of canine

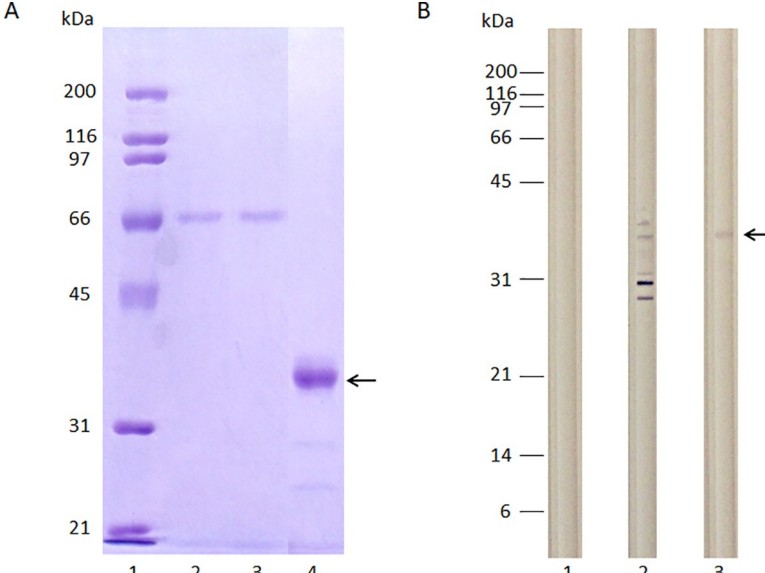

**Fig 2. Evaluation of purified rcasIL-10R1 by SDS–PAGE and Western blot.** RcasIL-10R1 was produced in High-five cells infected with the AcBacΔcc-GP64-casIL-10R1-6H baculovirus construct at MOI 5 for 72 h. Then, rcasIL-10R1 was purified from cell-free and virus-free culture supernatant (SN) by Ni-Sepharose affinity chromatography column. Samples of purified protein were evaluated by SDS–PAGE (A): molecular weight markers (lane 1), cell culture SN applied to the chromatographic column (lane 2), flow through (lane 3), and purified protein (lane 4) or Western blot developed by anti-his antibodies (B): SN from cells infected with baculovirus devoid of insert (negative control) (lane 1), SN from cells infected with AcBacΔcc-GP64-casIL-10R1-6H baculovirus construct (lane 2) or sample of purified rcasIL-10R1. Arrows indicate a band around 42 kDa corresponding to rcasIL-10R1-6H.

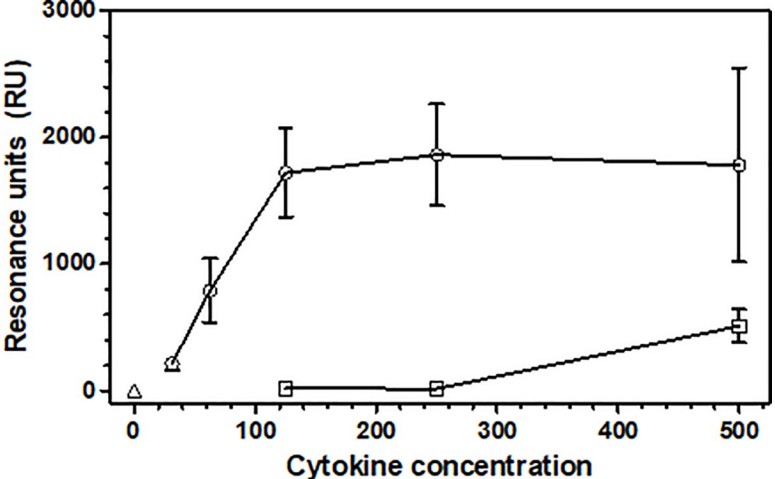

**Fig 3. Evaluation of binding between rcasIL-10R1 and canine IL-10.** Purified rcasIL-10R1 was immobilized on a CM5 chip in a Biacore T100 analyzer by applying 800 μL of the recombinant protein (0.5 μg/mL) at (50 μL/minute) to the chip matrix activated by 1-Ethyl-3-(3-dimethylaminopropyl) carbodiimide hydrochloride (EDC)/ N-Hydroxysuccinimide (NHS) to achieve 1000 resonance response units (RU). Then remaining reactive chemical groups on the chip were blocked by applying Ethanolamine hydrochloride-NaOH for one minute. The following samples were applied to the matrix for two minutes and 30 seconds: a) PBS containing 1% BSA and 0.05% Tween 20 (open triangle); b) canine IL-4 at 125, 250 or 500 ng/mL (open square); c) canine IL-10 at 31.2, 62.5, 125, 250 or 500 ng/ml (open circle). IL-4 and IL-10 were diluted with PBS containing 0.05% Tween 20. After each analyte binding evaluation, matrix regeneration was performed by applying regeneration buffer for 30 seconds. Each sample was evaluated twice and results are presented as means and standard deviations (X±SD) of RU.

IL-10 were applied, there was a progressive increase in signal starting at 211 ± 52 RU for 31.2 ng/mL and reaching a plateau at 1720 ± 352 RU for 125 ng/mL, indicating a strong binding between rcasIL-10R1 and IL-10 (Fig 3). Binding equilibrium constant ($EC_{80}$) between rcasIL-10R1 and IL-10 was determined as 51.4 nM.

### 3.3 RcasIL-10R1 inhibits proliferation of MC/9 cells stimulated with IL-4 and IL-10

Previously, Thompson-Snipes et al. (1991) [46], showed that MC/9 cells proliferate after dual stimulation with IL-10 and IL-4. To determine if rcasIL-10R1 was able to interfere with signaling by canine IL-10, MC/9 cells were cultured for 48 h in: a) complete DMEM (negative control) or complete DMEM with either b) 1.25% of supernatant from Con-A-stimulated rat splenocytes (positive control), c) rcaIL-4 (180 ng/mL), d) rcaIL-10 (20 ng/mL), e) rcaIL-4 180 ng/mL and rcaIL-10 at 20 ng/mL), and f) rcaIL-4 (180 ng/mL), rcaIL-10 (20 ng/mL) and rcasIL-10R1 (4 μg/mL). Then, Alamar Blue was added to cell cultures and, after 24 h, optical density (difference in measurements at 570 nm and 600 nm, $OD_{570-600nm}$), which correlates with the number of cells in wells, was determined. MC/9 cells cultured in medium alone or medium containing supernatant from Con-A-stimulated rat splenocytes revealed $OD_{570-600nm}$ values (X±SD) of 0.138 ± 0.014 and 0.465 ± 0.020, respectively (Fig 4). MC/9 cells stimulated with both rcaIL-4 and rcaIL-10 in the presence of rcasIL-10R1 exhibited lower $OD_{570-600nm}$ values (0.313 ± 0.039), as compared to cells activated in the absence rcasIL-10R1 (0.455 ± 0.042).

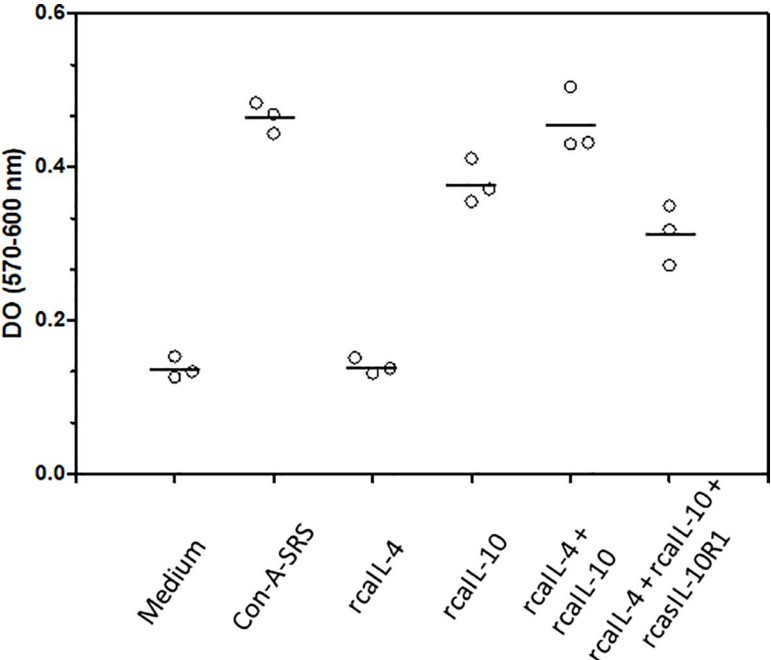

**Fig 4. Blocking canine IL-10 signaling by rcasIL-10R1 reduces MC/9 cell proliferation.** The murine mast cell line MC/9 was cultured at $1 \times 10^5$/mL in triplicate wells (100 μL/well) on a 96-well flat-bottomed microtiter plate with: a) complete DMEM alone (negative control) or complete DMEM containing: b) 0.625% of supernatant from Con-A-stimulated rat splenocytes (Con-A-SRS, assay positive control), c) rcaIL-4, 180 ng/mL, d) rcaIL-10, 20 ng/mL, e) rcaIL-4 (180 ng/mL) and rcaIL-10 (20 ng/mL), or f) rcaIL-4 (180 ng/mL), rcaIL-10 (20 ng/mL) and rcasIL-10R1 (4 μg/mL). The plate was kept for 48 h under 5% $CO_2$ at 37°C. Then, 10 μL of Alamar Blue were added to each well. Cells were cultured for an additional 24 hours and optical density (OD) was read at 570 nm and 600 nm wavelengths. Differences in mean OD values were used to estimate MC/9 cell proliferation rates. Symbols and bars represent replicates and means. RasIL-10R1 inhibited proliferation of MC/9 cells stimulated with IL-10 and IL-4 by 45%.

RcasIL-10R1 inhibited proliferation of MC/9 cells by 45%, as determined after subtracting background $OD_{570-600nm}$ value (observed from cells cultured in medium alone, 0.138) from values obtained when either IL-10 and IL-4 or IL-10, IL-4 and rcasIL-10R1 were added to cultures. Thus, suggesting that rcasIL-10R1 partially inhibited cell proliferation by blocking IL-10 signaling.

### 3.4 RcasIL-10R1 induces peripheral blood lymphocyte proliferation in dogs with leishmaniasis caused by *Leishmania infantum*

Dogs with leishmaniasis exhibit limited specific-cellular immune response and increase in IL-10 production [34,47], to determine if blocking IL-10 signaling would revert *Leishmania*-specific lymphoproliferative unresponsiveness, CFSE labeled-PBMCs from healthy or infected dogs were cultured together with, or without, rcasIL-10R1, and with or without the addition of SLA, or in the presence of PHA alone for five days. The Mean Fluorescence Intensities (MFI) of CFSE-labeled lymphocytes was determined under each condition. Reductions in CFSE-fluorescence were considered an indicator of cell proliferation [48]. The data described herein was previously reported in the context of testing combinations of several recombinant canine proteins [45]. In healthy dogs, lymphoproliferation was observed when PBMCs were cultured with PHA (median, interquartile 25, and 75, 256, 176, and 337) (Fig 5A), as compared with medium alone (101, 82, and 225). In diseased dogs, although CFSE-labeled lymphocytes cultured with PHA showed reductions in MFI, these were not statistically significant (Fig 5B). Lymphocytes from diseased dogs showed proliferative response when cultured with rcasIL-10R1, regardless of the addition of SLA to cultures (without SLA addition, rcasIL-10R1: 2.9, 2.0, and 10.2 vs medium: 128, 117, and 205; with SLA addition, rcasIL-10R1: 3.8; 1.3, and 12.1 vs medium: 121, 87, and 176) (Fig 5B). *L. infantum* parasites are transported throughout blood circulation mainly inside mononuclear leucocytes [49] and these parasites were detected in the blood samples of every single diseased dog by real-time PCR in the current studies. One probable explanation for lymphoproliferation stimulation by rcasIL-10R1, regardless of the addition of SLA to the cell cultures of diseased dogs, was the presence of *Leishmania* within the PBMCs used in experimentation. These results suggest that blocking IL-10 signaling using rcasIL-10R1 restores specific lymphoproliferative response in dogs with leishmaniasis.

## 4 Discussion

IL-10 can restrict exaggerated inflammatory and immune responses, thus preventing tissue damage and promoting homoeostasis [5,6,9]. However, by downregulating these responses, IL-10 may favor the development and/or persistence of chronic infections [50–52]. Therefore, blocking IL-10 signaling may contribute to the establishment of adequate immune responses for the treatment of chronic infections [37,50,53]. Blocking IL-10 signaling can also be useful in immunization protocols that aim to induce immune responses against intracellular pathogens [50,54].

IL-10 signaling can be blocked *in vitro* by the use of IL-10 or IL-10R reactive molecules, including antibodies, oligonucleotide or peptide aptamers, as well as soluble IL-10 receptor [39,41,53,55–57]. However, to date, only blocking antibodies to canine IL-10 or IL-10R have been developed (https://www.rndsystems.com R&D Systems), [38]. Such antibodies were produced in mice or goats, therefore, their administration in dogs could result in humoral responses to heterologous proteins [58], that would limit their use as blocking agents to IL-10 signaling in these animals.

The present work endeavored to identify the canine casIL-10R1 amino acid sequence, generate a recombinant baculovirus chromosome encoding this molecule, which was expressed in

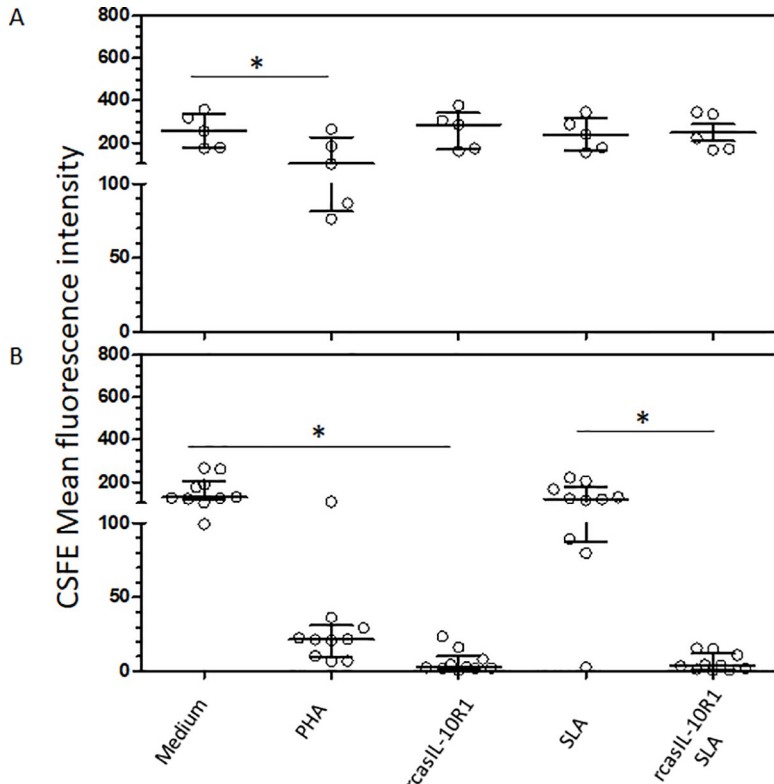

**Fig 5. Blocking canine IL-10 signaling by rcasIL-10R1 restores *in vitro* specific lymphoproliferative response in dogs with VL.** CFSE-labeled PBMCs from healthy negative control dogs (n = 5) (A) and dogs with leishmaniasis (n = 10) (B) were cultured in medium alone (Medium), medium with soluble leishmania antigens (SLA) or phytohemagglutinin (PHA). In addition, PBMCs cultured in medium alone or with SLA were stimulated with rcasIL-10R1. After 5 days, the mean fluorescence intensity (MFI) of CFSE-labeled lymphocytes was assessed by flow cytometry. Bars represent MFI median values and 25th and 75th percentile interquartile range. Symbols represent data from individual animals. Asterisks indicate significant differences (Friedman's test with Dunn's multiple comparison, p < 0.05).

insect cells and subsequently purified to obtain rcasIL-10R1. In addition, rcasIL-10R1 was evaluated *in vitro* with respect to its binding ability and blocking of the homologous IL-10 signaling pathway, as well as promoting lymphoproliferation in dogs with leishmaniasis caused by *L. infantum*.

Initially, casIL-10R1 was identified by comparing the amino acid sequences of caIL-10R1, huIL-10R1, husIL-10R1, and then detecting the extracellular domains in the first two proteins using an online tool for transmembrane domain prediction. Next, a DNA construct encoding casIL-10R1 was synthesized and transferred to a baculovirus artificial chromosome used to produce the protein in High five cells. Chromatographic affinity protein purification from cell culture supernatants indicated an adequate yield, reaching 2.8 mg/L. The generated recombinant protein presented a high degree of purity, as evidenced by a main band of 42 kDa when evaluated on SDS-PAGE and Western blotting assays. Since the predicted molecular weight of rcasIL-10R1 was 28 kDa, and considering its six canonical N-linked glycosylation motives (NXS/T), it follows that the protein must have been produced in a heavily glycosylated form. Similarly, a discrepancy was noted between the predicted molecular weight (24 kDa) and relative mobility (35–45 kDa) of rhusIL-10R1 produced in myeloma cells on SDS-PAGE analysis [41]. Moreover, these authors reported that treatment with N-glycanase promoted a reduction in molecular weight back to 24 kDa, indicating that the produced rhusIL-10R1 was highly

glycosylated [41]. In addition, our analysis of the purified protein in solution presented a low concentration of endotoxin [59].

To assess its binding ability, rcasIL-10R1 was immobilized on a dextran matrix and resonance was recorded following the application of different concentrations of canine IL-10. In comparison to diluent alone or canine IL-4 (negative control), much higher resonance values were observed for IL-10, indicating specific binding between rcasIL-10R1 and IL-10. The equilibrium constant (EC80) concentration was determined to be 51.4 nM. In a previous report, the established equilibrium constant (EC50) for human IL-10 and hus-IL-10R1 binding was 0.47 nM [60], which is much lower than that found herein. These observed discrepancies can be at least partially attributed to divergencies in the experimental conditions used, including the use of EC80 and native dimeric IL-10 in our protocols in comparison to EC50 and a mutated monomeric protein, in addition to differences in the methods of immobilization employed.

In combination with IL-4 and/or IL-3, IL-10 has been shown to induce mast cell proliferation [46], as demonstrated by growth in the mouse mast cell line (MC/9) through the concomitant stimulation of homologous IL-4 and IL-10. Moreover, MC/9 cells and a subcloned line, so-called MC/9.2, which expresses a lower amount of growth factor mRNA [61], have been used by several authors and biotechnology companies to demonstrate the functional activity of IL-10 in many animal species [46,62–64], including *Canis familiaris* (https://www.rndsystems. com/ R&D systems catalog number 735-CL-010 data sheet). To test the ability of rcasIL-10R1 in the blockade of the cognate signaling pathway, MC/9 cells were stimulated with canine IL-10 and IL-4 in the presence or absence of rcasIL-10R1 (4 µg/mL). In the presence of the recombinant protein, an incomplete reduction was observed in the proliferation of MC/9 cells, indicating the partial blocking of this signaling pathway. Tan et al. (1995) [41] showed that 15–20 nM of rhusIL-10R1 induced a 50% inhibition in the maximal proliferation of Ba8.1 cells (murine pro-B lymphocytes transfected with the gene encoding huIL-10R1) under stimulation with human IL-10 at 100 pM. In the present study, rcasIL-10R1 (95 nM) was found to promote a 45% reduction in the proliferation of MC/9 cells stimulated with 6 nM of canine IL-10.

Dogs naturally infected with *L. infantum* that remain asymptomatic have been shown to mount a specific lymphoproliferative response. However, dogs that succumb to the disease evolve with T cell exhaustion, involving both CD4+ and CD8+ T lymphocytes [34,47], which implies the loss of these cells' ability to perform effector functions. One of the first functions lost due to this exhaustion is the capacity of lymphocytes to proliferate intensely under antigenic stimulation [65]. In the current study, the blocking of IL-10 signaling with rcasIL-10R1 for 5 days in infected canine PBMCs, under stimulation or not by SLA, resulted in the restoration of a lymphoproliferative response. Since *Leishmania* DNA was detected in the peripheral blood of these dogs, the observed lymphoproliferation was quite likely specific. By contrast, Esch et al., (2013) [47] carried out assays in PBMCs from dogs with leishmaniasis caused by *L. infantum* to evaluate the impact of blocking IL-10 signaling with anti-IL-10 antibodies. In these assays, the authors assessed the percentage of T CD4 or T CD8 lymphocytes that incorporated EdU (5-ethynyl-2'-deoxyuridine) at 7 days of culture after stimulation with leishmania antigens in the presence of anti-IL-10 antibodies or an isotype control. They observed no increases in the percentages of either CD4 or CD8 T lymphocytes after the blockade of IL-10 signaling, suggesting the absence of lymphocyte proliferation. The discrepancies between these authors' results and those herein likely occurred due to differences in the methodology used to evaluate lymphocyte proliferation. In consonance with our results, the blocking of this signaling cascade with anti-IL-10 antibodies in PBMCs from human patients with visceral leishmaniasis using a method similar to that employed herein was also shown to result in lymphoproliferation [66].

Future investigations designed to determine the conditions in which rcasIL-10R1 would block IL-10 in vivo in a wider context will be of great scientific interest, and could be applied to induce a Th1 immune response in the development of vaccines and immunotherapeutic protocols against chronic infection and cancer in dogs.

## 5 Conclusion

The rcasIL-10R1 produced in this baculovirus-insect cell system demonstrated the blockade of the IL-10 signaling pathway and the restoration of *in vitro* lymphoproliferative response in dogs with leishmaniasis caused by *L. infantum*.

## Supporting information

**S1 Raw images.**
(PDF)

## Acknowledgments

The authors would like to thank Elivani Sacramento for helping with the experiments. The authors are also grateful to the Program for Technological Development of Tools for Health (PDTIS-FIOCRUZ) for the use of its facilities and to the Postgraduate Course of Biotechnology in Health and Investigative Medicine.

## Author Contributions

**Conceptualization:** Valeria Marçal Felix de Lima, Geraldo Gileno de Sá Oliveira.

**Formal analysis:** Catiule de Oliveira Santos, Sidnei Ferro Costa, Fabiana Santana Souza, Jessica Mariane Ferreira Mendes, Cristiane Garboggini Melo de Pinheiro, Diogo Rodrigo de Magalhães Moreira, Valeria Marçal Felix de Lima, Geraldo Gileno de Sá Oliveira.

**Funding acquisition:** Valeria Marçal Felix de Lima, Geraldo Gileno de Sá Oliveira.

**Investigation:** Catiule de Oliveira Santos, Sidnei Ferro Costa, Cristiane Garboggini Melo de Pinheiro, Diogo Rodrigo de Magalhães Moreira, Luciano Kalabric Silva, Valeria Marçal Felix de Lima.

**Methodology:** Catiule de Oliveira Santos, Sidnei Ferro Costa, Jessica Mariane Ferreira Mendes, Cristiane Garboggini Melo de Pinheiro, Diogo Rodrigo de Magalhães Moreira, Luciano Kalabric Silva.

**Project administration:** Valeria Marçal Felix de Lima, Geraldo Gileno de Sá Oliveira.

**Software:** Luciano Kalabric Silva.

**Supervision:** Cristiane Garboggini Melo de Pinheiro, Valeria Marçal Felix de Lima, Geraldo Gileno de Sá Oliveira.

**Writing – original draft:** Catiule de Oliveira Santos, Sidnei Ferro Costa, Diogo Rodrigo de Magalhães Moreira.

**Writing – review & editing:** Sidnei Ferro Costa, Fabiana Santana Souza, Jessica Mariane Ferreira Mendes, Valeria Marçal Felix de Lima, Geraldo Gileno de Sá Oliveira.

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
