## [Decision Letter · Decision Letter 0]

21 Oct 2020

PONE-D-20-26981

Blocking IL-10 signaling with soluble IL-10 receptor restores specific lymphoproliferative response in dogs with leishmaniasis caused by Leishmania infantum

PLOS ONE

Dear Dr. Oliveira,

Thank you for submitting your manuscript to PLOS ONE. After careful consideration, we feel that it has merit but does not fully meet PLOS ONE’s publication criteria as it currently stands. Therefore, we invite you to submit a revised version of the manuscript that addresses the points raised during the review process.

We look forward to receiving your revised manuscript.

Kind regards,

Yara M. Traub-Csekö

Academic Editor

PLOS ONE

Journal Requirements:

3. We noted in your submission details that a portion of your manuscript may have been presented or published elsewhere:

'Some data on lymphocyte proliferation has been previously published, however, detailed information on cloning and in vitro evaluation of soluble canine IL-10 receptor was no included in the previous publication.'

Please clarify whether this publication was peer-reviewed and formally published.

If this work was previously peer-reviewed and published, in the cover letter please provide the reason that this work does not constitute dual publication and should be included in the current manuscript.

Reviewers' comments:

Reviewer's Responses to Questions

**Comments to the Author**

1. Is the manuscript technically sound, and do the data support the conclusions?

Reviewer #1: Yes

Reviewer #2: Yes

2. Has the statistical analysis been performed appropriately and rigorously? 

Reviewer #1: Yes

Reviewer #2: No

3. Have the authors made all data underlying the findings in their manuscript fully available?

Reviewer #1: Yes

Reviewer #2: Yes

4. Is the manuscript presented in an intelligible fashion and written in standard English?

Reviewer #1: Yes

Reviewer #2: Yes

5. Review Comments to the Author

Reviewer #1: The authors identified and produced a recombinant canine sIL-10R1 and used that protein to block IL-10 signaling in vitro. Blocking of IL-10 signaling was done in vitro in PBMC obtained from infected and healthy / uninfected dogs. The work is interesting and can be considered for publication in Plosone, however some points must be clarified.

• Please modify the title to make it clear that this is an in vitro study. As it is, it makes us think that some treatment was done in vivo on the animals. In this sense, also modify the discussion and conclusion sections.

• In the abstract, define the abbreviation CanL as canine leishmaniasis, as it should be written (line 25).

• In relation to healthy animals, the authors should consider that they cannot exclude the possibility of Leishmania infection only by the result of blood qPCR. Some authors have shown in the literature that the load / determination of the parasite in the blood of infected animals can fluctuate during infection, sometimes showing negative results. Authors should associate the method used to identify parasites in the blood with serological methods such as DPP Leshmaniose or Elisa as currently described in the literature.

• In lines 77-78, the authors described that “Dogs that develop the symptomatic form of leishmaniasis exhibit higher IL-10 and lower IFN-γ concentrations in the blood, while the inverse it true in asymptomatic dogs [34, 35]”. However, the authors must consider that these results are controversial in the literature. Some authors have demonstrated a reduction in the expression of pro and anti-inflammatory cytokines in symptomatic dogs, including IL-10.

• The figure 1 caption looks confused. Please indicate only the letters A and B as shown in the figure.

• How was the statistical analysis of the data represented in figure 4?

• Figure 4 shows the effect of blocking IL-10 signaling in the presence of recombinant IL-10 and IL-4 on mast cell proliferation. Why wasn't the test also performed using only recombinant IL-10 and casIL-10R? Blocking this signaling appears to have had little effect on mast cell proliferation.

• In the caption of figure 5, please indicate that the recovery of the lymphoproliferative response was detected in vitro using cells obtained from dogs. Thus, it does not appear that any treatment has been done on animals in vivo.

• In figure 5, include the letters A and B for the upper graph and the lower graph, respectively.

• The authors wrote in lines 68-70 that “IL-10 also inhibits some T lymphocyte functions indirectly through decreased antigen presentation [24] and directly by inhibiting CD4 T cell proliferation and cytokine production (IL-2 and IFN-y) [25, 26]”, and in lines 277-280 that “Dogs with leishmaniasis exhibit limited specific-cellular immune response and increase in IL-10 production [34, 47], to determine if blocking IL-10 signaling would revert Leishmania- 9 specific lymphoproliferative unresponsiveness, CFSE labeled-PBMCs from healthy or infected dogs were cultured together with, or without, rcasIL-10R1, and with or without the addition of SLA, or in the presence of PHA alone for five days.” In this context, how could the authors explain that cells from healthy donors stimulated with PHA proliferated less than cells from sick dogs? As shown in figure 5.

• In relation to the above comment, it would be interesting to measure the production of IL-10 from the culture supernatant and dog plasma. It would also be interesting to measure the production of IFN-g and TNF after blocking IL-10 signaling.

Reviewer #2: It is an interesting paper, which propose the obtaining of a recombinant canine soluble IL-10R1 receptor using bacmid expression in insect cells and that this recombinant protein could restore in vitro lymphocytes proliferation in response to antigen from dogs naturally infected with Leishmania. However some points should be revised by the authors:

Line 40 – aggression replace by damage

Line 74 lower case for chagasi

Line 230. The authors could provide the comparison of amino acid sequence of human and canine IL-10 receptor R1

Line 266-272 and Figure 4 legends - The results description should be rewritten considering the percentage of inhibition as well as the authors could show the data as % of inhibition. It is better visualized.

Other point is about the experiments: what was the n of experiments ? one experiemtn in triplicates, two ??? and the statistical analysis ? Is there any statistical test applied ? It is not described in the graphic and legend.

Figure 5- Please, identify the graphics as A and B . It is quite confused this data , the graph below seems to show the proliferation of lymphocytes from disease dogs, and PHA induce cells a significant profliferation compared to medium alone, since a reduction on CFSE MFI is observed. Therefore the lines 289-287 describe “In diseased dogs, although CFSE-labeled lymphocytes cultured with PHA showed reductions in MFI, these were not statistically significant (Fig 1B).

Line 287 - Fig1B should be Fig 5B

Line 288 - These data should be better explained : “ Lymphocytes from diseased dogs showed proliferative response when cultured with rcasIL-10R1, regardless of the addition of SLA to cultures (without SLA addition, rcasIL-10R1: 2.9, 2.0, and 10.2 vs medium: 128, 117, and 205; with SLA addition, rcasIL-10R1: 3.8; 1.3, and 12.1 vs medium: 121, 91 87, and 176) (Fig 5B).”

Sincelery.

6. PLOS authors have the option to publish the peer review history of their article (what does this mean?). If published, this will include your full peer review and any attached files.

Reviewer #1: **Yes: **Fernanda Morgado

Reviewer #2: No

---

## [Author Response · Author response to Decision Letter 0]

10 Dec 2020

We would like to thank the reviewers for the questions and comments on the paper. For sure, after changes suggested by the reviewers, our paper has greatly improved.

Please find below the reviewers’ questions and comments on the paper and our answers to them.

Reviewer #1

Reviewer #1 comment

Please modify the title to make it clear that this is an in vitro study. As it is, it makes us think that some treatment was done in vivo on the animals. In this sense, also modify the discussion and conclusion sections.

Answer

Changes were carried out accordingly the reviewer suggestion.

Lines 2-3: Title: “Blocking IL-10 signaling with soluble IL-10 receptor restores specific lymphoproliferative response in dogs with leishmaniasis caused by Leishmania infantum” were changed to “Blocking IL-10 signaling with soluble IL-10 receptor restores in vitro specific lymphoproliferative response in dogs with leishmaniasis caused by Leishmania infantum”

Lines 320-32: Discussion: “In addition, rcasIL-10R1 was evaluated with respect to its binding ability and blocking of the homologous IL-10 signaling pathway, as well as promoting lymphoproliferation in dogs with leishmaniasis caused by L. infantum” were changed to “In addition, rcasIL-10R1 was evaluated in vitro with respect to its binding ability and blocking of the homologous IL-10 signaling pathway, as well as promoting lymphoproliferation in dogs with leishmaniasis caused by L. infantum.

Lines 392-394: Conclusion: “The rcasIL-10R1 produced in this baculovirus-insect cell system demonstrated the blockade of the IL-10 signaling pathway and the restoration of a lymphoproliferative response in dogs with leishmaniasis caused by L. infantum” were changed to “The rcasIL-10R1 produced in this baculovirus-insect cell system demonstrated the blockade of the IL-10 signaling pathway and the restoration of in vitro lymphoproliferative response in dogs with leishmaniasis caused by L. infantum.

Reviewer #1 comment 

In the abstract, define the abbreviation CanL as canine leishmaniasis, as it should be written (line 25).

Answer

Lines 25-27: Abstract: “Dogs that succumb to leishmaniasis caused by L. infantum (CanL) develop exhaustion of T lymphocytes and are unable to mount appropriate cellular immune responses to control the infection” were changed to “Dogs that succumb to canine leishmaniasis (CanL) caused by L. infantum develop exhaustion of T lymphocytes and are unable to mount appropriate cellular immune responses to control the infection”

Reviewer #1 comment 

In relation to healthy animals, the authors should consider that they cannot exclude the possibility of Leishmania infection only by the result of blood qPCR. Some authors have shown in the literature that the load / determination of the parasite in the blood of infected animals can fluctuate during infection, sometimes showing negative results. Authors should associate the method used to identify parasites in the blood with serological methods such as DPP Leshmaniose or Elisa as currently described in the literature.

Answer

In fact, the healthy control dogs were also evaluated by ELISA and showed no Leishmania-specific antibodies. 

In lines 185-188: The text “All control dogs tested negative for Leishmania DNA by real-time PCR, and presented complete blood counts and mean serum biochemistry parameters within reference ranges [45].” was changed to “All control dogs tested negative for Leishmania DNA and Leishmania-specific antibodies by real-time PCR and ELISA, respectively, and presented complete blood counts and mean serum biochemistry parameters within reference ranges [45].”

Reviewer #1 comment

In lines 77-78, the authors described that “Dogs that develop the symptomatic form of leishmaniasis exhibit higher IL-10 and lower IFN-γ concentrations in the blood, while the inverse it true in asymptomatic dogs [34, 35]”. However, the authors must consider that these results are controversial in the literature. Some authors have demonstrated a reduction in the expression of pro and anti-inflammatory cytokines in symptomatic dogs, including IL-10.

Answer

In fact, Luna et al. Vet. Immunol. Immunopathol. 70(1-2):95-103, 1999, presented data which suggest that dogs infected with L. infantum experience an immunosuppression state. 

Lines 77-78: “Dogs that develop the symptomatic form of leishmaniasis exhibit higher IL-10 and lower IFN-γ concentrations in the blood, while the inverse it true in asymptomatic dogs [34, 35]. ” were changed to “Dogs that develop the symptomatic form of leishmaniasis may exhibit higher IL-10 and lower IFN-γ concentrations in the blood, while the inverse it true in asymptomatic dogs [34, 35]”.

Reviewer #1 comment

The figure 1 caption looks confused. Please indicate only the letters A and B as shown in the figure.

Answer

Lines 613-622: Figure 1 caption was change to: “Fig. 1 Identification and design of DNA construct encoding soluble canine IL-10R1 production. Identification of the extra-cytoplasmic domain of canine IL-10 receptor alpha chain was performed by comparing the amino acid sequences of human (huIL-10R1, GeneBank, accession number NM_001558) and canine (caIL-10R1, Genbank accession number XM_005620306.1) IL-10 receptor alpha chain, and husIL-10R1, previously described by Tan et al. 1991, using Basic Local Align Search Tool, as well as defining signal peptide and transmembrane domains (TM) (defined by vertical bars) using an online tool (https://tmdas.bioinfo.se/DAS/) (A). The DNA construct was designed to encode the following elements in tandem: a) Autographa californica multiple nuclear polyhedrosis virus (AcMNPV) GP64 leader sequence, b) casIL-10R1, and c) six histidines (B).”

Reviewer #1 question

How was the statistical analysis of the data represented in figure 4?

Answer

To assess in vitro blocking of IL-10 signaling by rcasIL-10R1, a murine mast cell line was used (MC/9 cells). The assay was carried out in triplicate of wells. The data shown in figure 4 was evaluated only by descriptive statistics. OD means of triplicates were determine and used to compare different culture conditions. Tan, J. C., et al. J Biol Chem 270(21): 12906-12911, 1995, who described the characterization of soluble human IL-10 receptor used the cell line Ba8.1 and show only descriptive statistics.

Reviewer #1 comment and question

Figure 4 shows the effect of blocking IL-10 signaling in the presence of recombinant IL-10 and IL-4 on mast cell proliferation. Why wasn't the test also performed using only recombinant IL-10 and casIL-10R? Blocking this signaling appears to have had little effect on mast cell proliferation.

Thompson-Snipes, L., et al. J Exp Med 173(2): 507-510,1991, described that murine IL-10 on its own was unable to promote proliferation of murine mast cell line MC/9 cells. Murine IL-10 was able to induce proliferation of MC/9 cells only in the presence of murine IL-3 or IL-4. It was a little bit surprising that, in our experiments, canine IL-10 by itself was able to induce proliferation of MC/9 cells. The reviewer is right to point out that blocking canine IL-10 stimulation in MC/9 cells did not have a considerable effect on MC/9 cells proliferation induced by the combination of canine IL4 and IL-10, although rcasIL-10R1 could bind canine IL-10 (see figure 3) and stimulate proliferation of peripheral blood lymphocytes from dogs with CanL (figure 5).

Reviewer #1 comment

In the caption of figure 5, please indicate that the recovery of the lymphoproliferative response was detected in vitro using cells obtained from dogs. Thus, it does not appear that any treatment has been done on animals in vivo.

Answer

Lines 660-661: Caption of figure 5 was changed from “Fig 5. Blocking canine IL-10 signaling by rcasIL-10R1 restores specific lymphoproliferative response in dogs with VL.” to “Fig 5. Blocking canine IL-10 signaling by rcasIL-10R1 restores in vitro specific lymphoproliferative response in dogs with VL.”

Reviewer #1 comment

In figure 5, include the letters A and B for the upper graph and the lower graph, respectively.

Answer

In figure 5, letters A and B were included in the upper graph and the lower graph, respectively.

Reviewer #1 comment

The authors wrote in lines 68-70 that “IL-10 also inhibits some T lymphocyte functions indirectly through decreased antigen presentation [24] and directly by inhibiting CD4 T cell proliferation and cytokine production (IL-2 and IFN-y) [25, 26]”, and in lines 277-280 that “Dogs with leishmaniasis exhibit limited specific-cellular immune response and increase in IL-10 production [34, 47], to determine if blocking IL-10 signaling would revert Leishmania- 9 specific lymphoproliferative unresponsiveness, CFSE labeled-PBMCs from healthy or infected dogs were cultured together with, or without, rcasIL-10R1, and with or without the addition of SLA, or in the presence of PHA alone for five days.” In this context, how could the authors explain that cells from healthy donors stimulated with PHA proliferated less than cells from sick dogs? As shown in figure 5.

Answer

Before comparing the effect of different culture conditions, a normality statistical test (Shapiro-Wilk test) was used and the data shown in figure 5 revealed a non-normal distribution. Therefore, comparison between cells subjected to different treatments was performed by the Friedman test and comparison between each two treatments was made by the Dunn´s test. In the control dogs, Friedman test showed a statistically significant difference between MFI medians of cells subjected to different treatments. In addition, the Dunn´s test showed a statistically significant difference between MFI median of cells cultivated with culture medium alone and cells cultivated in the presence of PHA (see figure 5). For dogs with CanL, Friedman's and Dunn's tests were also performed. The latter showed no statistically significant difference between the MFI median of cells cultivated with culture medium alone and cells cultured in the presence of PHA (see figure 5). This was may be related to variance variabilities.

Reviewer #1 comment

In relation to the above comment, it would be interesting to measure the production of IL-10 from the culture supernatant and dog plasma. It would also be interesting to measure the production of IFN-g and TNF after blocking IL-10 signaling.

Answer

The reviewer is right. Unfortunately, IL-10 was not measure in plasma from dogs in the current study. IFN-g, TNF-a and IL-10 were measured in culture supernatants, however, their concentrations were low and there was no statistically significant different in the concentration of these cytokines (medium alone vs medium plus rcasIL-10R1) in either control or infected dogs.

Reviewer #2

Reviewer #2 comment

Line 40 – aggression replace by damage

Answer

Line 40: The word “aggression” was replaced by “damage”.

Reviewer #2 comment

Line 74 lower case for chagasi

Answer

Line 74: The word “chagasi” was rewritten in lower case.

Reviewer #2 comment

Line 230. The authors could provide the comparison of amino acid sequence of human and canine IL-10 receptor R1

Answer

Lines 231-232: It is pointed out that “CasIL-10R1 was defined as an array of 215 amino acids which exhibited 74% similarity to homologous region of human protein”.

Reviewer #2 comment

Line 266-272 and Figure 4 legends - The results description should be rewritten considering the percentage of inhibition as well as the authors could show the data as % of inhibition. It is better visualized.

Answer

Lines 270-276: The text was rewritten to indicate the percentage of proliferation inhibition. “MC/9 cells stimulated with both rcaIL-4 and rcaIL-10 in the presence of rcasIL-10R1 exhibited lower OD570-600nm values (0.313 ± 0.039), as compared to cells activated in the absence rcasIL-10R1 (0.455 ± 0.042), suggesting that rcasIL-10R1 partially inhibited cell proliferation by blocking IL-10 signaling.” Was replaced by “MC/9 cells stimulated with both rcaIL-4 and rcaIL-10 in the presence of rcasIL-10R1 exhibited lower OD570-600nm values (0.313 ± 0.039), as compared to cells activated in the absence rcasIL-10R1 (0.455 ± 0.042). RcasIL-10R1 inhibited proliferation of MC/9 cells by 45%, as determined after subtracting background OD570-600nm value (observed from cells cultured in medium alone, 0.138) from values obtained when either IL-10 and IL-4 or IL-10, IL-4 and rcasIL-10R1 were added to cultures. Thus, suggesting that rcasIL-10R1 partially inhibited cell proliferation by blocking IL-10 signaling.”

Lines 363-365: It was rewritten “In the present study, rcasIL-10R1 (95 nM) was found to promote a 45% reduction in the proliferation of MC/9 cells stimulated with 6 nM of canine IL-10.”

Lines 657-658: It was rewritten “RasIL-10R1 inhibited proliferation of MC/9 cells stimulated with IL-10 and IL-4 by 45%.”

Reviewer #2 comment and questions

Other point is about the experiments: what was the n of experiments ? one experiemtn in triplicates, two ??? and the statistical analysis ? Is there any statistical test applied ? It is not described in the graphic and legend.

Answer

Line 152: it is indicated that binding analysis of rcasIL-10R1 to IL-10 was carried out twice.

Line 164: it is indicated that murine mast cell (MC/9 cell) proliferation was performed in triplicate of wells.

Lines 215 and 667: it is indicated that analysis of lymphoproliferative response was carried out by Friedman’s test followed by Dunn’s test.

Reviewer #2 comment

Figure 5- Please, identify the graphics as A and B . It is quite confused this data , the graph below seems to show the proliferation of lymphocytes from disease dogs, and PHA induce cells a significant profliferation compared to medium alone, since a reduction on CFSE MFI is observed. Therefore the lines 289-287 describe “In diseased dogs, although CFSE-labeled lymphocytes cultured with PHA showed reductions in MFI, these were not statistically significant (Fig 1B).

Answer

Figure 5, the upper graph and the lower graph were labeled “A” and “B”, respectively.

Initially, before comparing different conditions of cell culture, a normality statistical test (Shapiro-Wilk test) was used and the data, shown in figure 5, revealed a non-normal distribution. Therefore, comparison between cells subjected to different treatments was performed by the Friedman test and comparison between each two treatments was made by the Dunn´s test. In the control dogs, Friedman test showed a statistically significant difference between MFI medians of cells subjected to different treatments. In addition, the Dunn´s test showed a statistically significant difference between MFI median of cells cultivated with culture medium alone and cells cultivated in the presence of PHA (see figure 5). For dogs with CanL, Friedman's and Dunn's tests were also performed. The latter showed no statistically significant difference between the MFI median of cells cultivated with culture medium alone and cells cultured in the presence of PHA (see figure 5). This was may be related to variance variabilities.

Reviewer #2 comment

Line 287 - Fig1B should be Fig 5B

Answer

Line 291: “Fig1B” was replaced by “Fig 5B”

Reviewer #2 comment

Line 288 - These data should be better explained : “ Lymphocytes from diseased dogs showed proliferative response when cultured with rcasIL-10R1, regardless of the addition of SLA to cultures (without SLA addition, rcasIL-10R1: 2.9, 2.0, and 10.2 vs medium: 128, 117, and 205; with SLA addition, rcasIL-10R1: 3.8; 1.3, and 12.1 vs medium: 121, 91 87, and 176) (Fig 5B).”

Answer

L. infantum parasites are transported throughout blood circulation mainly inside mononuclear leucocytes (Paraguai de Souza, E., et al. Acta Trop 80(1): 69-75, 2001) and these parasites were detected in the blood samples of every single diseased dog by real-time PCR in the current studies. One probable explanation for lymphoproliferation stimulation by rcasIL-10R1, regardless of the addition of SLA to the cell cultures of diseased dogs, was the presence of Leishmania within the PBMCs used in experimentation. This explanation was added in lines 295-300.

---

## [Decision Letter · Decision Letter 1]

2 Jan 2021

Blocking IL-10 signaling with soluble IL-10 receptor restores specific lymphoproliferative response in dogs with leishmaniasis caused by Leishmania infantum

PONE-D-20-26981R1

Dear Dr. Oliveira,

We’re pleased to inform you that your manuscript has been judged scientifically suitable for publication and will be formally accepted for publication once it meets all outstanding technical requirements.

Kind regards,

Yara M. Traub-Csekö

Academic Editor

PLOS ONE

Additional Editor Comments (optional):

Reviewers' comments:

Reviewer's Responses to Questions

**Comments to the Author**

1. If the authors have adequately addressed your comments raised in a previous round of review and you feel that this manuscript is now acceptable for publication, you may indicate that here to bypass the “Comments to the Author” section, enter your conflict of interest statement in the “Confidential to Editor” section, and submit your "Accept" recommendation.

Reviewer #1: All comments have been addressed

2. Is the manuscript technically sound, and do the data support the conclusions?

Reviewer #1: Yes

3. Has the statistical analysis been performed appropriately and rigorously? 

Reviewer #1: Yes

4. Have the authors made all data underlying the findings in their manuscript fully available?

Reviewer #1: Yes

5. Is the manuscript presented in an intelligible fashion and written in standard English?

Reviewer #1: Yes

6. Review Comments to the Author

Reviewer #1: (No Response)

7. PLOS authors have the option to publish the peer review history of their article (what does this mean?). If published, this will include your full peer review and any attached files.

Reviewer #1: No

---

## [Editor Report · Acceptance letter]

8 Jan 2021

PONE-D-20-26981R1 

Blocking IL-10 signaling with soluble IL-10 receptor restores *in vitro* specific lymphoproliferative response in dogs with leishmaniasis caused by *Leishmania infantum*

Dear Dr. de Sá Oliveira:

I'm pleased to inform you that your manuscript has been deemed suitable for publication in PLOS ONE. Congratulations! Your manuscript is now with our production department. 

Kind regards, 

on behalf of

Dr. Yara M. Traub-Csekö 

Academic Editor

PLOS ONE